

# Brief Communication: Does it matter exactly when the Arctic will become ice-free?

J. K. Ridley, R. A. Wood, A. B. Keen, E. Blockley, J. A. Lowe

Met Office Hadley Centre, FitzRoy Road, Exeter EX1 3PB, UK

*Correspondence to:* J. K. Ridley (jeff.ridley@metoffice.gov.uk)

**Abstract.** Following the 2015 UNFCCC Conference of Parties in Paris there is renewed interest in understanding and avoiding potentially dangerous climate change. The loss of Arctic sea ice is one of the most directly visible aspects of climate change and the question is frequently asked: when can we expect the Arctic to be ice-free in summer? We argue here that this question may not be the most useful one to inform decisions on climate change mitigation or adaptation in the Arctic. The development of a community-wide consensus on a robust definition of 'ice-free', may reduce confusion in the community and amongst the public.

## 1. Introduction

Arctic sea ice is an iconic feature of the climate system, and its annual retreat to its minimum extent in September provides a regular checkpoint on Arctic climate change. Since 1979 there has been a decline of around 30% in the summer minimum ice extent (Stroeve et al., 2012). However the long term decline is modulated by strong year-to-year variability. The record low extent in 2012 was followed by a strong recovery in 2013 and 2014 (NSIDC, 2015), accompanied by an increase in the volume of the ice (Tilling et al., 2015), a combination of a cold winter and strong ice convergence to the Canadian Archipelago (Kwok et al., 2015) forming the thick ice .One might then speculation that the Arctic ice cover is more sensitive to fluctuations in temperature than previously thought.

In 2015 the minimum ice extent was reached on 11th September and was 4.41 million km2 (NSIDC, 2015), substantially lower than 2013 and 2014 and close to the long term declining trend line since the start of the modern satellite record in 1979. The 2015 melt season was notable for high rates of melting during July, which is generally thought to be caused by anti-cyclonic conditions over Greenland and the central Arctic (Matsumura et al., 2014); this emphasises the important role played by synoptic weather conditions in the year-to-year variations in sea ice. Overall, the internal variability of the Arctic sea ice (Swart et al., 2015) leads to no clear evidence of either a recent slowdown or a speeding-up of the longer term trend of sea ice loss.

The long-term decline of Arctic sea ice that has been observed since 1979 is robustly projected by climate models to continue over the coming decades (Collins et al., 2013), and will have substantial impacts on regional ecosystems, indigenous peoples, transport and resource exploitation (Larsen et al.,2014). However, because of internal variability, the appearance of ice-free conditions on a particular date may be of limited significance, depending on the stakeholder. For example the opening of navigation routes for just a few days in an individual year would be of limited value, while particular ecosystems (especially organisms with multi-year life cycles) may be robust to short term variations. It may be more important to know when and where sustained ice-free conditions can be expected.



## 2. Method & Results

We consider four plausible definitions of the date of an 'ice-free Arctic'. We apply the commonly-used threshold, for which northern hemispheric sea ice extent (defined as the total area of ocean with a sea ice fraction greater than 15%) is less than 1 million km2, as the definition of the term 'ice-free'. The threshold of 1 million km2, rather than zero, is used because ice can be expected to remain for some time along the northern coast of Greenland, whilst for navigational purposes the central Arctic is ice-free. The 'first ice-free year' is then defined as:

A. The first year that at least one day is 'ice free'.

B. The first year when the September mean is 'ice free'.

C. The first time the final year of a 5 year running mean of September monthly mean extents is 'ice-free'.

D. The final year of 5 consecutive September monthly means which are 'ice-free'.

A and B are point definitions which, when passed, are likely to evoke significant public interest. However the point at which the threshold is crossed is likely to be strongly dependent on natural year-to-year variability as well as the long-term trend. C and D are attempts to smooth out the effects of interannual variability to produce a more robust measure. Criteria B-D have all been used in the literature and in the IPCC 5th Assessment (Collins et al., 2013). By definition, $A \leq B < C \leq D$. Other, more impact-specific criteria are possible but not considered here.

We assess the impact of the different definitions in small (4-member) ensembles of climate model projections using the climate model HadGEM2-ES (Martin et al., 2011), for the representative greenhouse gas concentration pathways, RCP4.5 and RCP8.5 (Van Vuuren et al., 2011). Each ensemble member is taken to represent a plausible realisation of the future Arctic and our focus is on the effect of the choice of definitions on the declared 'ice-free date', within each realisation.

The dates for which the HadGEM2-ES ensemble depicts the first year with an ice free day range between 2032-2042 for RCP8.5 and 2038-2046 for RCP4.5 (Figure 1). The lag between this first day (criterion A) and the progressively stricter criteria B-D is indicated by a progression of the ice-free dates for each ensemble member. The difference in declared ice-free date between criteria A and D is between 4 and 19 years across all the ensemble members.

The ensemble spread in ice-free date, for a given definition of ice-free conditions (an indicator of the importance of interannual variability) is similar in both scenarios and all ice-free definitions, at 8 ± 1 years, except for RCP4.5 criterion D. Thus, the uncertainty in the date on which the Arctic might be declared ice-free may depend as much on the specific criterion for 'ice-free' as on the internal variability or the RCP scenario.

## 3. Discussion

Our ensembles are too small to sample the full range of interannual variability, and they come from a single climate model. However they are sufficient to illustrate that the choice of definition could have a substantial impact on the date when an 'ice-free' Arctic is declared. In particular we see that the 'AR5' definition D, which might be expected to reduce the effects of interannual variations, can actually amplify them. In RCP4.5 this is because the long-term declining trend becomes quite slow near the threshold (compared with the inter-annual variability), resulting in many crossings and re-crossings.





75 The evidence of summer 2015 is that the Arctic is continuing its progress towards seasonally ice-free conditions.

76 Many of the impacts of decreasing ice cover will be felt irrespective of the precise date when the Arctic is

77 declared seasonally ice-free. Further, both natural systems and human activities will need to operate in an

78 environment of strong year-to-year variability. Recent progress in seasonal forecasting has shown promise that

79 reliable predictions of the year-to-year variations may be possible (Peterson et al., 2014). To inform climate

80 change mitigation and adaptation to the effects of the changing Arctic climate it may be more important to

81 develop seasonal-to-decadal sea-ice predictions, alongside impact-relevant metrics of ice loss, rather than

82 refining projections of the ice-free date.

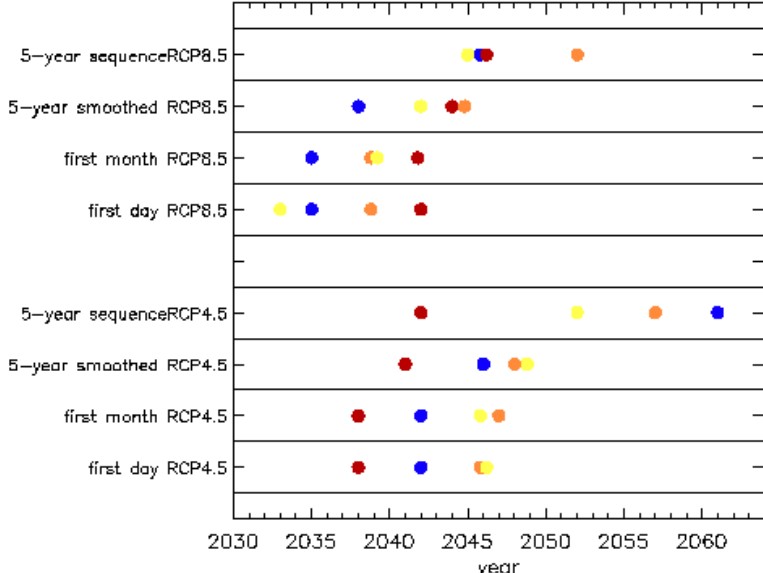

83

84 Figure 1. The dates of ice-free Arctic from various definitions (y-axis) for each of four HadGEM2-ES ensemble

85 members (colours) for the RCP4.5 (lower) and RCP8.5 (upper) scenarios.

86

87 **References**

88 Collins, M., R. Knutti, J. Arblaster, J.-L. Dufresne, T. Fichefet, P. Friedlingstein, X. Gao, W.J. Gutowski, T.

89 Johns, G. Krinner, M. Shongwe, C. Tebaldi, A.J. Weaver and M. Wehner, 2013: Long-term Climate Change:

90 Projections, Commitments and Irreversibility. In: Climate Change 2013: The Physical Science Basis.

91 Contribution of Working Group I to the Fifth Assessment Report of the Intergovernmental Panel on Climate

92 Change [Stocker, T.F., D. Qin, G.-K. Plattner, M. Tignor, S.K. Allen, J. Boschung, A. Nauels, Y. Xia, V. Bex

93 and P.M. Midgley (eds.)]. Cambridge University Press, Cambridge, United Kingdom and New York, NY, USA

94

95 Kwok, R. (2015), Sea ice convergence along the Arctic coasts of Greenland and the Canadian Arctic

96 Archipelago: Variability and extremes (1992–2014), Geophys. Res. Lett., 42, 7598–7605,

97 doi:10.1002/2015GL065462.



Larsen, J.N., O.A. Anisimov, A. Constable, A.B. Hollowed, N. Maynard, P. Prestrud, T.D. Prowse, and J.M.R.
Stone, 2014: Polar regions. In: Climate Change 2014: Impacts, Adaptation, and Vulnerability. Part B: Regional
Aspects. Contribution of Working Group II to the Fifth Assessment Report of the Intergovernmental Panel on
Climate Change [Barros, V.R., C.B. Field, D.J. Dokken, M.D. Mastrandrea, K.J. Mach, T.E. Bilir, M.
Chatterjee, K.L. Ebi, Y.O. Estrada, R.C. Genova, B. Girma, E.S. Kissel, A.N. Levy, S. MacCracken, P.R.
Mastrandrea, and L.L. White (eds.)]. Cambridge University Press, Cambridge, United Kingdom and New York,
NY, USA, pp. 1567-1612.

Martin, G.M., N. Bellouin, W. J. Collins, I. D. Culverwell, P. R. Halloran, S. C. Hardiman, T. J. Hinton, C. D.
Jones, A. J. McLaren, F. M. O'Connor, et al. 2011 The HadGEM2 family of Met Office Unified Model Climate
Configurations, Geosci. Model Dev., 4, 723-75.  DOI: 10.5194/gmd-4-723-2011

NSIDC National Snow and Ice Data Center, 2015:  http://nsidc.org/news/newsroom/2015-arctic-sea-ice-
minimum

Matsumura, S.,  Zhang, X.and Yamazaki K., 2014: Summer Arctic Atmospheric Circulation Response to Spring
Eurasian Snow Cover and Its Possible Linkage to Accelerated Sea Ice Decrease. J. Climate, 27, 6551–6558.doi:
http://dx.doi.org/10.1175/JCLI-D-13-00549.1

Stroeve, J. C., Kattsov, V., Barrett, A., Serreze, M., Pavlova, T., Holland, M. and Meier, W.N., 2012 Trends in
Arctic sea ice extent from CMIP5, CMIP3 and observations, Geophys. Res. Lett., 39, L16502,
doi:10.1029/2012GL052676.

Swart, N. C., J. C. Fyfe, E. Hawkins, J. E. Kay      and  A.  Jahn, 2015: Influence of internal variability on
Arctic sea-ice trends, Nature Climate Change, 5, 86−89  doi:10.1038/nclimate2483

Tilling, R.L., A. Ridout, A. Shepherd, D.J. Wingham 2015.  Increased Arctic sea ice volume after anomalously
low melting in 2013, Nature Geosci., 8, doi:10.1038/ngeo2489

Peterson, K. A., Arribas, A., Hewitt, H. T., Keen, A. B., Lea, D. J. and McLaren, A. J., 2014, Assessing the
forecast skill of Arctic sea ice extent in the GloSea4 seasonal prediction system, Clim. Dynam., 1−16,
doi:10.1007/s00382-014-2190-9

Van Vuuren DP, Edmonds J, Kainuma M et al (2011) The representative concentration pathways: an overview.
Clim Chang 109:5−31. doi:10.1007/s10584-011-0148-z