# Peer review of "Brief Communication: Does it matter exactly when the Arctic 2 will become ice-free?"

_The Cryosphere, 2016_

## Referee Comment (RC1) · Anonymous Referee #1 · 4 Apr 2016

In this very short contribution, the authors present results from 4 ensemble members of the HadGEM2-ES for the RCP8.5 and RCP4.5, and ask the question how different definitions of "first ice free year" affect the answer within this small ensemble, as well as the more general question as to how useful an answer to this question is. And while it is interesting to see a comparison of the impact of 4 different definitions of "first ice-free year" in a small ensemble of 4 simulations from one model, the result is not particularly surprising. The second point of the contribution, as reflected in the title, is the question whether we, as sea ice community, should focus on when the Arctic will be ice-free for the first time. And while I completely agree that the crossing of an arbitrary threshold (such as 1 million km2) has little real world meaning, it is a fact that the crossing of such thresholds (as the recent rise above 400 ppm $CO_2$ in the atmosphere, discussions about avoiding more than 2 C global warming, etc) is

something than captures attention in the media and the public. I do agree, however, that a common definition of how we define "first year of ice-free" as a community would be very worthwhile.

Finally, I disagree strongly with the final paragraph of the conclusion of the article (lines 75-82), which suggests that we have reliable seasonal forecasting systems that can predict year-to-year sea ice variability, so that we should focus our energy there, rather than on refining sea ice projections from climate models. There remains a large spread in seasonal forecasts of the sea ice extent (see the SIPN Sea ice Outlook for last year, https://www.arcus.org/sipn/sea-ice-outlook/2015/post-season), with so far limited skill. And with a range of over 100 years in the prediction of the "first ice-free year" in the CMIP5 models, and a large range in general in the projections of sea ice in CMIP5 (temporarily and spatially) I would argue that refining climate model projections of sea ice is still a top priority in order to allow more reliable adaptation planning, and narrow down when in the 21st century we can expect the Arctic to become ice-free. I would end after line 74, as that is really all that can be said based on the results presented.

Overall the article reads like a nice poster written up into an article, with some issues that could easily be fixed. However, I am concerned that this contribution is really below the "least-publishable unit" threshold, even for a "Brief communication", but I will leave that decision up to the editor.

Specific minor comments: Line 17: this sentence reads strangely, seems like a "result of" is missing between the end of line 17 and the start of line 18. Otherwise it is unclear how line 18 links up with line 17 logically. Line 19: Completely unclear how the authors come up with this speculation that "the sea ice cover is more sensitive to fluctuations in temperature than previously thought", as they even explicitly mention the impact of strong sea ice convergence in the line above. I would recommend removing this statement, also because the author do not come back to this at all, and it is not supported by any data or linked to any of the results or conclusions in the paper. Line 33-34: It is unclear to me what the "while particular ecosystems may be robust to short

term variations" mean here and what the purpose of this statement is. Please rephrase. Line 46: I would suggest rephrasing this definition of C, as it is confusing as it is. What is meant, I think, is the first time a 5 year running mean of September monthly mean extents is "ice-free". The "final year of" is confusing here, and not needed if we are looking at a running mean. Or maybe I misunderstood completely, but that also shows it should be rephrased or explained better.

―――――――――――――――――――

---

## Referee Comment (RC2) · Anonymous Referee #2 · 28 Apr 2016

Brief Communication: Does it matter exactly when the Arctic will become ice-free? By J. K. Ridley, R. A. Wood, A. B. Keen, E. Blockley, J. A. Lowe

This short communication defends the idea that existing definitions of ice-free Arctic may be insufficient to inform stakeholders and societies about impacts of climate change in the Arctic and decisions for climate change mitigation, and that more robust definitions would be welcome.

I am worried by two points. First, current definitions of "ice-free" conditions are indeed varying greatly from study to study, but I do not see that as a problem. Indeed, these definitions are meant to represent more a symbolic (and sad) milestone in the evolution of climate change than an exact threshold beyond which the system will undergo, instantaneously, drastic transformations. In the same idea, why did stakeholders

choose to target emissions to stay below a 2°C above present? This threshold is also arbitrary: an anomaly of 2°C is equivalent to 3.6° Fahrenheit, which is not a round number. Should the UN have adopted another metric system, the target could have been slightly different (for example, 4°F to have a round number, which is 2.2°C). These types of milestones are meant to give a gross idea of limits not to exceed, not stringent limits that should be interpreted literally. That different definitions give different results is therefore not surprising.

Second, no real alternative is proposed by the authors. This note would have been very constructive if it provided new diagnostics encapsulating spatial information, information about uncertainty related to internal variability and to model error as well as to RCP scenario. This is not the case here.

In summary, I am concerned by the fact that this short notes addresses more a wording issue than a true scientific question, and by the fact that it does not resolve the question it is raising in the introduction. This makes me think that The Cryosphere might not be the best place to get this note published.

Other comments:

Line 19 - Delete blank space after ice and include one before "One".

Line 19 - "One might then speculation" –> "One might then speculate".

line 32: I don't understand the sentence: "For example the opening of navigation routes for just a few days in an individual year would be of limited value, while particular ecosystems (especially organisms with multi-year life cycles) may be robust to short term variations". What is meant by "robust to short term variations"?

Line 45 - "B. The first year when the September mean is 'ice free'.". In that definition, do the authors use the "September mean of daily sea ice extent", or the "Sea ice extent computed from the September mean of daily sea ice concentration"? The two are different since extent is a threshold definition.

[Figure]

Fig. 1 is of very poor graphical quality. In addition, can the authors label the four definitions with letters A, B, C, D as in the text? Finally, I have the impression that for RCP4.5 the year of ice-free according to the definition "first day" (last row) for the yellow member occurs *later* than the year of ice-free conditions according to the definition "first month" (while it cannot be the case by construction). That is, the two yellow dots on the two last rows of the figure are not aligned as expected. Is that just an optical effect?

---

## Editor Comment (EC1) · J. Stroeve (Editor) · 2 May 2016

In regards to this brief communication, I am in agreement with the reviewers comments that not much new knowledge is gained in this brief communication. If the intent is more of a suggestion to the sea ice community to use consistent language when talking about ice-free conditions, then such a discussion is better suited for a journal such as EOS, which publishes opinion pieces such as this (see recent EOS article about What Darkens the Greenland Ice Sheet (https://eos.org/opinions/what-darkens-the-greenland-ice-sheet).

Thus, I feel that unless major revisions are made as per the reviewers comments this communication is not suitable for publication in The Cryosphere